# Divergent Impact of Enzyme Replacement Therapy on Human Cardiomyocytes and Enterocytes Affected by Fabry Disease: Correlation with Mannose-6-phosphate Receptor Expression

**DOI:** 10.3390/jcm11051344

**Published:** 2022-02-28

**Authors:** Andrea Frustaci, Behzad Najafian, Giuseppe Donato, Romina Verardo, Cristina Chimenti, Luigi Sansone, Manuel Belli, Enza Vernucci, Matteo Antonio Russo

**Affiliations:** 1Department of Clinical, Internal, Anesthesiologist and Cardiovascular Sciences, La Sapienza University, 00100 Rome, Italy; cristinachimenti@libero.it (C.C.); enza.vernucci@gmail.com (E.V.); 2Cellular and Molecular Cardiology Laboratory, IRCCS L. Spallanzani, 00149 Rome, Italy; romina.verardo@inmi.it; 3Department of Pathology, University of Washington, Seattle, WA 98195, USA; najafian@uw.edu; 4Department of Translational and Precision Medicine, La Sapienza University, 00100 Rome, Italy; giuseppe.donato@uniroma1.it; 5Laboratory of Molecular and Cellular Pathology, IRCCS San Raffaele Pisana, 88163 Rome, Italy; luigi.sansone@sanraffaele.it (L.S.); manuel.belli@sanraffaele.it (M.B.); 6MEBIC Consortium, San Raffaele Open University, 00166 Rome, Italy; matteoantoniorusso44@gmail.com; 7IRCCS San Raffaele Pisana, 88163 Rome, Italy

**Keywords:** Fabry disease, GLA, globotriaosylceramide, cardiomyocyte, intestine, enzyme replacement therapy, molecular and cellular rehabilitation

## Abstract

Background: The impact of enzyme replacement therapy (ERT) on cardiomyocytes and intestinal cells, affected by Fabry disease (FD), is still unclear. Methods: Six patients with FD, including five family members with GLA mutation c.666delC and one with GLA mutation c.658C > T, manifesting cardiomyopathy and intestinal symptoms (abdominal pain, diarrhea and malabsorption) were included in the study. Clinical outcome, cardiac magnetic resonance (CMR), endomyocardial and gastro-intestinal biopsies were evaluated before and after 2 years of treatment with agalsidase-α (0.2 mg/kg every other week). Immunohistochemistry and Western blot assessments of mannose-6-phosphate receptors (IGF-II-R) on intestinal and myocardial frozen tissue were obtained at diagnosis and after 2 years of ERT. Results: After ERT left ventricular maximal wall thickness, ranging from pre (<10.5 mm) to mild (<15 mm) and moderate hypertrophy (16 mm), was not associated with significant changes at CMR. Degree of dyspnea, mean cardiomyocyte diameter and % vacuolated areas of cardiomyocytes, representing intracellular GL3, remained unmodified. In contrast, intestinal symptoms improved with disappearance of diarrhea, recovery of anemia and weight gain, correlating with near complete clearance of the enterocytes from GL3 inclusions. IGF-II-R expression was remarkably higher even at histochemistry in intestinal tissue compared with myocardium (*p* < 0.001) either at baseline and after ERT, thus justifying intestinal recovery. Conclusions: Human cells affected by FD may respond differently to ERT: while cardiomyocytes retain their GL3 content after 2 years of treatment, gastro-intestinal cells show GL3 removal with recovery of function. This divergent response may be related to differences in cellular turnover, as well as tissue IGF-II-R expression.

## 1. Introduction

Fabry disease (FD) is an X-linked inborn error of glycosphingolipid catabolism that is caused by deleterious mutations in the GLA gene encoding the lysosomal hydrolase, alpha-galactosidase A (α-Gal A) [1,2]. The marked deficiency or absence of α-Gal A activity results in the systemic accumulation of globotriaosylceramide (GL3) and related glycosphingolipids within the lysosomes in various cells throughout the body, leading to a wide range of symptoms and complications [3,4,5,6,7]. Gastrointestinal involvement, although suggested in the first description by Joahnnes Fabry [8] and William Anderson [9], has been under-recognized for a long time. Indeed, gastrointestinal symptoms, including abdominal pain, diarrhea/constipation and intestinal malabsorption, have been reported in 19 to 70% of patient with Fabry disease in the literature [10,11,12]. These symptoms are linked to accumulation of GL-3 in intestinal epithelial cells, other stromal cells vessels and ganglion cells (Meissner’s and Auerbach’s plexus).

Currently, available treatments for FD are limited and include ERT (agalsidase-α and agalsidase-β since 2001) and chaperone therapy (Galactose since 2001 [13] and migalastat [14] since 2016), and a few novel approaches which are under investigation.

The efficacy of ERT on GL-3 clearance is cell-type specific and dose-dependent [15,16,17,18,19]. While 1 mg/kg every other week (EOW) of agalsidase-β clears endothelial and mesangial cells and fibroblasts from GL-3 inclusions in 5 months, podocytes show partial clearance after one year of treatment in males with classic FD. This discrepancy may be linked to different cellular life spans and turnover, since podocytes are post-mitotic/slow turnover cells as opposed to other kidney cell types. Cardiomyocytes, similar to podocytes, are post-mitotic and have long lifespans [20]. Our knowledge on the tissue effects of ERT on cardiomyocytes is rather scarce. Using a semi-quantitative approach, Thurberg et al. reported GL-3 clearance in cardiac endothelial cells after 5 months of agalsidase-β (1 mg/kg EOW), but no clearance in cardiomyocytes. Here, for the first time, we provide histologic data on agalsidase-α efficacy in cardiomyocytes and enterocytes after 2 years of ERT. Moreover, in order to examine if cellular lifespan is associated with different expression of mannose-6-phospate receptors (IGF-II-R), we evaluated if IGF-II-R in human myocardial and intestinal cells provided slow and rapid turnover, respectively, for comparison.

## 2. Materials and Methods

### 2.1. Clinical Characteristics of Patients

Six out of 88 patients with Fabry disease, with endomyocardial and intestinal biopsies, clinical and imaging studies before and 2 years after initiation of agalsidase-α were enrolled into this study. Clinical, imaging and biopsy findings of the patients are summarized in Table 1. Diagnosis of Fabry disease was established by determination of blood α- galactosidase A enzyme activity and genotyping of *GLA*. Five (M/F = 2/3) of the patients, age 21–72 years, were relatives with *GLA* gene mutation c.666delC. The sixth patient was a 35-year-old non-related man with *GLA* mutation c.658C > T. All patients presented with cardiomyopathy and palpitation and/or dyspnea, abdominal pain and diarrhea, and 2 patients had intestinal malabsorption (weight loss and iron-deficiency anemia).

In all patients, cardiac investigations included non-invasive (ECG, Holter, 2D-echo, CMR and invasive (Coronary, left ventricular angiography and endomyocardial biopsy) studies. In particular, ECG, Holter and 2D-echo were obtained every three months, while CMR and left ventricular endomyocardial biopsy were undertaken, after written informed consent, before and after 2 years of ERT (agalsidase alpha 0.2 mg/kg infused every other week) administration.

Gastric and duodenal biopsies were performed before and after 2 years of ERT.

Laboratory tests, including hemoglobin and serum iron determination were acquired in addition to the assessment of body weight before and after ERT as part of evaluations for intestinal malabsorption.

### 2.2. Cardiac Magnetic Resonance (CMR)

CMR exams were performed on a 1.5 Tesla scanner. Standard cardiac magnetic resonance protocol included (1) cine magnetic resonance images acquired during breath-holds in the short-axis, 2-chamber and 4-chamber; (2) black blood T2-weighted short tau inversion recovery images on short-axis planes covering the entire left ventricle during 6 to 8 consecutive breath-holds for myocardial edema detection; and (3) late gadolinium-enhanced imaging performed 15 min after injection of 0.2 nmol/kg of gadoteratemeglumine and signal intensity value 2 SDs above the mean signal intensity of the remote normal myocardium was considered suggestive for myocardial fibrosis. (Appendix A).

### 2.3. Invasive and Endomyocardial Biopsy Studies

Cardiac catheterization with left ventricular and coronary angiography was obtained in all patients. Endomyocardial biopsy (four to 8 samples each patient) was performed in the septal-apical region of left ventricle. 

### 2.4. Histology and Electron Microscopy

For histological analysis, the endomyocardial and gastric and duodenal biopsy samples were fixed in 10% buffered formalin and paraffin embedded. Five microns–thick sections were stained with hematoxylin and eosin and Masson trichrome. For electron microscopy studies, additional biopsy samples were fixed in 2% glutaraldehyde in a 0.1 M phosphate buffer, at pH 7.3, post-fixed in osmium tetroxide and processed by following a standard protocol for embedding in Epon resin. Ultrathin sections were stained with uranyl acetate substitute and lead hydroxide and analyzed by a JEOL-1400-plus electron microscope.

Assessment and comparison of mannose-6-phosphate receptors in the myocardial and intestinal tissues in Fabry disease patients:

To evaluate a possible receptor-mediated different ERT delivery in the myocardium and intestine, we determined the expression of mannose-6-phosphate receptor in frozen intestinal and myocardial tissue in all six patients before and after 2 years on ERT. Results were compared with values from normal myocardium and intestine.

### 2.5. Immunohistochemistry

The expression of mannose-6-phosphate receptors was evaluated by immunoperoxidase using IGF2R Recombinant Rabbit Monoclonal Antibody (SR45-09) (1:120, Invitrogen, Thermo Fisher Scientific 168 Third Avenue, Waltham, MA, USA). Intensity of immunostaining was semi-quantitatively evaluated as absent (grade 0), weak (grade 1), mild (grade 2) moderate (grade 3) or strong (grade 4). For each patient, the grading was calculated on 10 different histological sections, and the average value was computed.

### 2.6. Protein Isolation and Western Blot

Heart tissue samples were treated as previously described [10]. The expression of mannose-6-phosphate receptors was visualized by using IGF-II Receptor/CI-M6PR (D8Z3J) polyclonal antibody (1:1000, Cell Signaling Techology, Danvers, MA, USA). Anti-α-sarcomeric actin (1:500, Sigma-Aldrich) antibody was used for normalization. Signal was visualized by using a secondary horseradish peroxidase-labeled goat anti-mouse antibody (goat anti-mouse IgG-HRP 1:5000, SantaCruz Biotechnology, Dallas, TX, USA) and enhanced chemiluminescence (ECL Clarity Bio-Rad, Hercules, CA, USA). The purity, as well as equal loading (40γ) of the protein, was determined by Nanodrop One (Thermofisher, Waltham, MA, USA). To normalize target protein expression, the band intensity of each sample was determined by densitometry with Image J software. Next, the intensity of the target protein was divided by the intensity of the loading control protein. This calculation adjusts the expression of the protein of interest to a common scale and reduces the impact of sample-to-sample variation. Relative target protein expression can then be compared across all lanes to assess changes in target protein expression across samples. Digital images of the resulting bands were quantified by the Image Lab software package (Bio-Rad Laboratories, Munchen, Germany) and expressed as arbitrary densitometric units.

### 2.7. Morphometric Studies

Cardiomyocyte’s cross-sectional area was computed by measuring the cardiomyocyte diameter across the nucleus in 50 to 100 cells cut transversely. At that level, the total area of glycolipid bodies was measured, and the percent of cardiomyocyte total area occupied by glycolipid bodies was calculated. These measurements were analyzed by using Nis-Elements BR software.

## 3. Results

Common morphological (OM and TEM) features in Fabry disease’s myocardial, gastric and duodenal biopsy samples before and after ERT.

(see Appendix A).

### 3.1. Myocardium 

OM shows perinuclear and intracardiomyocyte bodies, which, at TEM, appeared as osmiophilic myelin-like glycolipid bodies (MLGLBs), typically observed in Fabry diseases.

Myelin-like glycolipid storage bodies are membrane-bound vacuoles containing electron-dense membranaceous material, which, at higher magnification, appear organized similarly to the myelin membranes.

Before and after ERT, as evaluated by morphometry, there are no significant differences in the glycolipid bodies content. See Figure 1, Figure 2, Figure 3 and Figure 4.

### 3.2. Gastric Mucosa

Before ERT, mucosal epithelium is apparently unaffected at OM. At TEM occasional typical osmiophilic myelin-like glycolipid bodies of limited extend may be observed in muciparous cells of the epithelium. Endocrine cells are also unaffected.

ML-GLBs are mostly observed in sub epithelial mesenchymal cells, mostly fibroblasts, macrophages and endothelial cells. 

After ERT, in all patients studied, glycolipid bodies are absent, both at OM in thick sections and at TEM in ultrathin sections. The limitation of this evaluation resides in the dimensional characteristics of samples which have a small surface area and a limited thickness, excluding the deep layers of the submucosa. See Figure 5.

### 3.3. Duodenal Mucosa

Before ERT, mucosal epithelium is apparently unaffected both at OM and TEM. Endocrine cells are also unaffected. See Figure 6.

Subepithelial mesenchimal (mostly fibroblasts, macrophages, smooth muscle cells and endothelial cells) frequently contains typical osmiophilic myelin-like glycolipid bodies of limited extend.

After ERT, glycolipid bodies are no more visible. However, a number of autophagocytic vacuoles are present. They may contain mitochondria, membrane debris and small myelin-like glycolipid bodies; sometimes they are devoid of dense structures (ghost bodies), suggesting a previous digestion of GB3.

The clinical and pathologic characteristics of our 6 patients are summarized in Table 1.

### 3.4. Results of Mannose-6-phosphate Receptors Determination

#### 3.4.1. Immuno-Histochemistry Study

Semi-quantitative evaluation of immunostaining (grades from 0 to 4) for mannose-6-phosphate receptors showed an intense positive cytoplasmic immunostaining cardiomyocyte expression in normal heart (Figure 7A) compared with FD myocardial tissue (Figure 7B), (3.35 ± 0.5 vs. 0.34 ± 0.3) *p* = 0.002 (*p*  <  0.001), and a normal expression in control intestinal tissue versus FD intestine (Figure 7 panel C,D; 3 ± 0.25 vs. 2.72 ± 0.4), *p*= 0.1455 (NS).

#### 3.4.2. Western Blot Analysis

The protein analysis showed that mannose-6-phosphate receptor expression in normal intestinal tissue was 1.6-fold increased compared with normal myocardium (38,376 ± 1788 vs. 23,117 ± 2230, respectively, *p* < 0.01). Of note, in Fabry disease, intestinal tissue expression was 1.7-fold higher compared to normal bowel (63,784 ± 12,018 vs. 38,376 ± 1788, *p* < 0.01), while FD myocardial expression was 7-fold lower than in normal heart (3312 ± 1110 vs. 23,117 ± 2230, *p* < 0.01). Thus, FD intestinal expression was 19-fold higher than FD myocardium (63784 ± 12,018 vs. 3312 ± 1110, *p* < 0.01). After 2 years of ERT, there was no statistically significant difference in intestinal and myocardial receptors compared with baseline values. (Figure 7)

#### 3.4.3. Statistical Analysis

The statistical analysis was performed by the GraphPad Prism package, version 5.02 (Graphpad Software Inc., San Diego, CA, USA). The comparison between groups was performed with Mann–Whitney non-parametric test was used. A *p*-value less than 0.05 was considered statistically significant.

## 4. Case Studies

### 4.1. Case 1

A 72-year-old normotensive lady presented with a long history of palpitation, dyspnea on effort, abdominal pain and diarrhea. Routine laboratory tests were unremarkable. In the past, she had undergone multiple negative gastric and intestinal endoscopy and received a diagnosis of irritable colon. She was normotensive with sinus rhythm and a heart rate of 68 bts/min; PR was 140 ms, and QRS voltages were in the normal limits. Holter registered 8300 VEBs with polymorphic couplets and triplets. 2D-echo showed mild LV hypertrophy (maximal wall thickness (MWT) 13.5 mm) with preserved end-diastolic diameter (48 mm) and Ejection Fraction (53%) and normal valves. CMR indicated an area of LGE in the LV lateral free wall and a reduced T1 mapping (873–896 ms) suggesting glycolipids accumulation. She underwent a coronary angiography, which showed normal vessels and a left ventricular endomyocardial biopsy. Her histology showed large cardiomyocytes with perinuclear vacuoles that, on frozen sections, were positive for GL3 (Figure 1A–C). TEM showed the presence of perinuclear myelin-like glycolipid bodies (Figure 1E). A genetic study revealed GLA gene mutation c.666delC, confirming the diagnosis of FD. Plasma Lyso-Gb3 was elevated (2.3 nMol/L; laboratory normal < 1.13). Her gastrointestinal tract was examined via endoscopy, and gastric and duodenal biopsies were taken. Immunofluorescence microscopy revealed GL3 accumulation in the visceral wall (Figure 1G). The ultrastructural examination documented a number glycolipid bodies inside subepithelial and mesenchymal cells denoting FD intestinal involvement (Figure 1I). The patient received agalsidase alpha at the dosage of 0.2 mg/kg every other week for 2 years. During follow-up, she reported an attenuation and then disappearance of gastrointestinal symptoms. Dyspnea on effort persisted but ventricular ectopic beats reduced to 880/24 h with no more repetitive phenomena. After 2 years of treatment, a new CMR and gastric, duodenal and endomyocardial biopsies were repeated. While myocardial hypertrophy and reduced T1 Mapping remained unchanged, intestinal GL3 deposition was no more visible by immunofluorescence or TEM (Figure 1B–J).

### 4.2. Case 2

A 54-year-old normotensive female from same family, with GLA gene mutation (c.666delC) and increased Lyso-Gb3 (3.8 nMol/L; laboratory normal < 2.3 nMol/L), presented with asymptomatic mild LV hypertrophy by 2D-echo (LV-MWT 13 mm) and recurrent abdominal pain and diarrhea. ECG showed sinus rhythm with short PR (116 ms) and negative T wave in D1, aVL and V3-V6 leads. Her CMR was characterized by reduced T1 Mapping (875–897 ms) and a mild LGE area in the LV posterior wall. Coronary angiography was normal. The LV endomyocardial biopsy showed moderately hypertrophied cardiomyocytes (24.91 ± 8.24 µm,) with intracellular vacuoles (Figure 1A) and unaffected cells because of random X-inactivation.

Vacuoles were positive for GL3 by immunofluorescence microscopy (Figure 2C) and presenting at TEM characteristic myelin-like glycolipid bodies (Figure 2E). Gastric and duodenal biopsies documented prominent GL3 accumulation in the intestinal cells by immunofluorescence microscopy (Figure 2G) with characteristic Fabry inclusion features by ultrastructural examination (Figure 2I). The patient received agalsidase alpha 0.2 mg/kg every other week and presented a rapid improvement and then disappearance of abdominal symptoms. Periodic non-invasive assessments and repeating CMR, endomyocardial, gastric and duodenal biopsies were performed after a 2-year treatment. LV MWT, as well as LGE and T1 Mapping, remained unchanged at 2 years post-ERT (Appendix A). Endomyocardial biopsy studies showed no significant difference in cardiomyocyte diameter or vacuolar area on H&E stains, or intracellular GL3 accumulation (glycolipid bodies) by immunofluorescence or TEM between pre-ERT and 2-year post-ERT biopsies (Figure 2A–G). Conversely, intestinal biopsy confirmed complete tissue clearance from GL3 deposits at 2 years post-ERT (Figure 2H–J).

### 4.3. Case 3

A 43-year-old normotensive man belonging to the same family as Cases #1 and #2, with *GLA* mutation (c.666delC), 0% α-Gal-A activity and elevated Lyso-Gb3 (69.22 nMol/L; laboratory normal < 2.3), presented with ECG sinus rhythm, but short PR interval (110 ms), increased QRS voltages and moderate LV hypertrophy (MWT 16 mm at 2-D echo) with preserved end-diastolic diameter (50 mm) and EF (60%), associated with effort dyspnea. The major symptoms were, however, related to gastrointestinal tract consisting of diarrhea and weight loss: lately he reached the weight of 40 kg and developed iron-deficiency anemia (Hb 9.8 gr/dL, serum ferritin 13 ng/mL). He had a sinus rhythm of 74 bts/min with short PR (115 ms) and increased QRS voltages at ECG. CMR showed a reduced T1 Mapping (T1 874–883 ms) indicating glycolipid accumulation and LGE positive areas in the LV lateral wall denoting fibrosis. The patient underwent a coronary angiography that showed normal arteries. An LV endomyocardial biopsy documented hypertrophied cardiomyocytes with large intracellular vacuoles on H&E stain that were positive for GL3 by immunofluorescence microscopy and contained myelin-like glycolipid bodies at TEM (Figure 3A–F). Gastric and duodenal biopsies revealed extensive intracellular GL3 accumulation by immunofluorescence microscopy (Figure 3G) and Fabry inclusions on ultrastructural evaluation (Figure 3J). This duodenal biopsy also included intestinal ganglion cells which showed cytoplasmic vacuolization. He was treated with agalsidase alpha (0.2 mg/kg every other week), followed every three months by a physical examination, Hb and serum iron and 2D-echocardiogram. After 2 years of treatment, he had repeated CMR, endomyocardial and gastric and duodenal biopsies. While cardiac symptoms, CMR and cardiomyocyte parameters (Figure 3B,D,F vs. Figure 3A,C,E) remained unchanged, diarrhea disappeared soon and anemia and serum iron were recovered (Hb 14.5 gr/dL), and body weight was increased by 5 kg (becoming 45 kg). A follow-up intestinal biopsy showed complete tissue clearance from GL3 inclusions (glycolipid bodies) by both immunofluorescence (Figure 3H vs. Figure 3G) and TEM (Figure 3J vs. Figure 3I). 

### 4.4. Case 4

A 29-year-old normotensive man from same family as the above cases and with analogous *GLA* gene mutation (c.666delC), 0% α-Gal activity and elevated plasma Lyso-Gb3 (36.5 nMol/L; laboratory normal < 2.3), was admitted because of palpitations, abdominal pain, diarrhea and weight loss (body weight 41 kg on admission). His blood pressure and ECG were normal, including PR duration (182 ms) and QRS voltages. Holter monitoring registered 7621 ventricular ectopic beats with 48 couplets and 2 triplets. The 2D-echo documented normal cardiac dimensions (LV end-diastolic diameter 45 mm), contractility (LVEF 59%) and wall thickness (LV-MWT 9 mm). The CMR showed (Figure 4A–C) no areas of LGE, normal myocardial relaxation (T2 mapping 48 ms; normal < 50), and reduced T1 mapping (range between 950 and 820 ms, mean value 887, normal > 970), suggesting possible early GL3 accumulation. To clarify the cause of cardiac arrhythmias, after informed consent, a coronary angiography was performed that showed normal vessels. The LV endomyocardial biopsy revealed mild cardiomyocyte hypertrophy (cell diameter 21.63 ± 3.01 µm) with perinuclear vacuoles on H&E stain that were positive for GL3 by immunofluorescence microscopy and contained multi-lamellar inclusions by electron-microscopy (Figure 4). Due to diarrhea, weight loss and iron deficiency anemia (Hb 9.1 gr/dL), the patient underwent gastric and duodenal biopsies, which revealed extensive GL3 accumulation (glycolipid bodies) in the epithelial, endothelial and mesenchymal cells. He received ERT with agalsidase alpha (0.2 mg/kg every other week) and was followed up every 3 months with blood tests, Holter and 2D-echo. After 2 years of treatment, he underwent CMR, endomyocardial and intestinal biopsies. Cardiac arrhythmias were reduced to 2800 ventricular ectopic beats with no more repetitive phenomena. His CMR remained unmodified (Figure 4D–F) in terms of cardiac parameters, LGE and T1 mapping (886 ms). The endomyocardial biopsy showed no significant change in cardiomyocyte diameter (23.75 ± 2.87µm) with persistent cytoplasmic vacuolization on H&E stain (Figure 4B vs. Figure 4A). Intracellular GL3 accumulation was also persistently present through the immunofluorescence microscopy and TEM (Figure 4D vs. Figure 4C and Figure 4G vs. Figure 4F) In contrast, the intestinal biopsy showed no more intracellular GL3 accumulation by light, immunofluorescence or TEM (Figure 4H–J).

### 4.5. Case 5

A 21-year-old normotensive girl from same family as above, (*GLA* mutation c.666delC), with normal α-Gal-A activity and plasma Lyso-Gb3 concentration (2.0 nMol/L; normal < 2.3), was admitted because of palpitations and severe diarrhea (up to 12 bowel movements per day), causing hypotension and syncope. Routine laboratory tests, including serum electrolytes (K + 3.7 Meq/L), iron and Hb, were within normal limits. ECG was normal, but Holter registered 4200 ventricular ectopic beats with 32 couplets and 5 triplets. The echocardiogram showed normal cardiac dimensions (LV end-diastolic diameter 49 mm), contractility (LVEF 62%) and left ventricular wall thickness (LV-MWT 8 mm). The CMR showed normal T1 mapping (1020 ms) and absence of LGE areas. Coronary angiography and LV endomyocardial biopsy were performed after informed consent to investigate the origin of electrical instability. Coronary arteries were normal, but the myocardial histology showed mildly hypertrophied cardiomyocytes (20-micron transverse diameter at nuclear level) containing perinuclear vacuoles on H&E stain that were positive for GL3 by immunofluorescence microscopy and were composed of enlarged lysosomes filled with myelin-like glycolipid bodies by TEM. Gastric and duodenal biopsies showed extensive GL3 accumulation in the intestinal cells. The patient started on agalsidase alpha at a dosage of 0.2 mg/kg every other week. After a few infusions, her diarrhea, palpitations and syncope disappeared. Holter registered 280 ventricular ectopic beats in 24 h with no repetitive activity. After 2 years of ERT, she had control CMR, endomyocardial and intestinal biopsies. The CMR findings were normal; however, the biopsy studies did not reveal any significant change in cardiomyocyte diameter or vacuolization by light microscopy, or GL3 accumulation by immunofluorescence or electron-microscopy. In contrast, the intestinal biopsy after 2 years of ERT showed no residual GL3 accumulation by light microscopy, IF or TEM.

### 4.6. Case 6

A 35-year-old normotensive man with c.658C > T mutation of *GLA* and 0% alpha-galactosidase A activity in leukocytes, denoting a classic form of Fabry disease, was admitted because of chest discomfort and palpitations, hypertrophic cardiomyopathy of mild severity (left ventricular maximal wall thickness 14 mm), short PR interval at ECG and concomitant abdominal pain with diarrhea. The patient underwent CMR, denoting reduced values of T1 mapping to 950 ms in the absence of abnormal LGE signals.

The patient, after informed consent, underwent coronary angiography, LV endomyocardial biopsy, gastrointestinal endoscopy with gastric and duodenal biopsies. Histology and electron-microscopy confirmed Fabry cardiomyopathy with GL3 deposition in the intestinal cells. He received ERT in the form of agalsidase alpha at the dosage of 0.2 mg/kg intravenously every other week. His intestinal symptoms stopped after the second infusion, while his chest symptoms persisted. After 2 years, the patient was re-evaluated with CMR and intestinal and cardiac biopsy. While the CMR and cardiac pathologic findings were unmodified, the intestinal biopsy showed the absence of intracellular MLBs accumulation.

## 5. Discussion

This study, for the first time, provided a detailed histologic assessment of paired pre- and post-ERT intestinal and cardiac biopsies after 2 years of treatment with 0.2 mg/kg EOW agalsidase-α. In fact, to our knowledge, this is the first report of the effect of ERT on intestinal tissue GL3 accumulation in Fabry disease. Gastrointestinal symptoms, including abdominal pain, bloating, diarrhea, constipation, recurrent nausea and vomiting, are common among patients with Fabry disease and have been reported in up to 70% of males [21]. These symptoms are also common in females and children [11,22]. While the beneficial effects of ERT on gastrointestinal symptoms have been reported for either agalsidase-α or agalsidase-β [23,24], whether these benefits are associated with tissue clearance from GL3, or other phenomena, such as improvement of inflammation, is not known. Our study documents that 2 years of agalsidase-α EOW completely clears enterocytes from GL3 inclusions. On the other hand, gastrointestinal symptoms, such as diarrhea and abdominal pain, were improved or resolved relatively shortly after the initiation of ERT. Whether improvement of these symptoms could be attributed to the observed tissue clearance (i.e., GL3 clearance occurred earlier than 2 years) or was related to improved cellular function not necessarily or directly linked to GL3 clearance cannot be known from our study. Answering these questions will be critical in understanding the pathophysiology of Fabry disease gastrointestinal complications and treatment. Improvement of the gastrointestinal symptoms was most notable in male patients with 0% α-Gal A activity with progressive initial improvement and then disappearance (normally within 6 months) of abdominal pain, diarrhea and malabsorption. The latter manifesting was accompanied by normalization of serum iron and hemoglobin, as well as with an increase in body weight of up to 5 kg.

Long-term follow-up studies and registry data suggest that ERT, especially if started early, may reduce the occurrence of cardiac events [25,26,27,28]. On the other hand, the efficacy of ERT in clearing myocardium from GL3 remains controversial. Hughes et al. reported a 20% reduction in myocardial GL3, measured by HPLC, after 6 months of 0.2 mg/kg EOW agalsidase-α [29]. In contrast, Thurberg et al., while reporting complete clearance of cardiac endothelial cells, did not report any significant change in cardiomyocyte GL3 after 54 months of ERT with 1 mg/kg EOW agalsidase-β. Given the higher dose and longer duration of ERT in Thurberg’s study, the validity of HPLC data in Hughes et al. is questionable. All patients enrolled in our study received agalsidase-α with a dose similar to Hughes et al. and did not show any obvious GL3 clearance in cardiomyocytes on endomyocardial biopsies after 2 years of treatment. However, detection of partial clearance of cardiomyocytes from GL3 which may require unbiased quantitative morphometry at TEM studies similar to what was reported for podocytes in the kidney cannot be excluded. Reduction in myocardial mass has been reported and ascribed to ERT administration [30]; however, direct evidence of GL3 clearance from cardiac tissue has not been documented. If ERT does not clear GL3 from cardiomyocyte, the reported reduction of myocardial mass by 2D-echocardiography and/or CMR could be due to cardiomyocyte death and myocardial fibrosis, which is not a beneficial pathway.

Our study, for the first time, compares endomyocardial and intestinal biopsies obtained before and after 2 years of agalsidase-α administration and provides correlations with clinical evolution and non-invasive studies. Thus, while the clearance of GL3 inclusions was clearly visible in the intestinal epithelial cells, it could not be readily documented in cardiomyocytes that instead remain hypertrophied with prominent glycosphingolipid accumulation. Importantly, GL3 clearance from cardiomyocytes was not demonstrable even in patients with mild (Patients 1 and 2 and 6 with MWT < 15 mm) or pre-hypertrophic (Patients 4 and 5, MWT < 10.5 mm) cardiomyopathy where interstitial fibrosis was absent or very limited to interfere with access of infused enzyme to the cells. This novel observation raises important questions on the mechanism of action of ERT and on the opportunity to increase ERT dosage for the treatment of FD.

On the other hand, we did not observe any significant increase in cardiomyocyte diameter or vacuolated surface area during the two years of treatment; while our study did not include any untreated controls, these findings suggest that ERT may stabilize the cardiomyopathy at least in its early phase. The general belief is that ERT provides its major clinical benefits in early stages of Fabry disease and may prevent or attenuate disease progression, concepts that would advocate for early diagnosis and initiation of treatment. In our study, disease stabilization, while failing to affect dyspnea, was associated with attenuation of arrhythmia. Thus, the frequency and complexity of ventricular ectopic beats were reduced and repetitive activities were disappeared in our patients after ERT. We do not have, at the moment, a clear explanation for arrhythmias improvement despite persistence of intracellular Gb3. We can speculate that reduced interstitial glycolipids, particularly those around cells of conduction tissue, might have had a beneficial role.

Our study clearly demonstrates a heterogeneous response to ERT on human cardiomyocyte and enterocyte. This divergent response appears to be more cell dependent rather than organ dependent. Podocytes and arterial smooth muscle cells are more resistant to ERT compared with endothelial cells, mesangial cells or fibroblasts in the kidney. Arterial smooth muscle cells in the skin and cardiomyocytes in the heart show more resistance to ERT compared to endothelial cells in the same tissues. One could speculate different reasons for these differences. IGF-II-R is the major intracellular carrier of α-Gal A, and, thereby, it is likely to have a response to ERT and the removal of GL3 in Fabry cells. The current study documented that IGF-II-R expression in enterocytes, which exhibit complete GL3 clearance after ERT, is overexpressed, while it is remarkably reduced in cardiomyocytes presenting values 7-fold lower than normal controls (Figure 7). Interestingly, re-evaluation of IGF-II-R expression after 2 year of ERT showed unchanged intestinal and myocardial figures. This observation suggests that ERT is unlikely to affect IGF-II-R metabolism both in intestinal and myocardial tissue.

These results are supportive of a role for IGF-II-R expression in heterogeneous response of different cell types to ERT. Understanding pathways involved in IGF-II-R expression may offer insights for novel therapies targeting ERT-resistant cells, such as cardiomyocytes. In addition, the exogenous enzyme may not be easily accessible to podocytes, cardiomyocytes or smooth muscle cells, due to the barrier created by the extracellular matrix between these cells and the blood, as opposed to the endothelial cells. Cellular turn over may play a significant role in the observed clearance from GL3 following ERT, a concept that has not been examined. Cardiomyocytes and podocytes are both post-mitotic cells, and vascular smooth muscle cells have a longer lifespan than endothelial cells. The turnover of cardiomyocytes is estimated to be about 1% per year at age 20 year and 0.3% at age 75 [20], while intestinal cells are completely replaced in 4 or 5 days [31]. Najafian et al. [32] showed that GL3 accumulation in podocytes is progressive with age, while such a relationship is not present in endothelial or mesangial cells, a phenomenon that may well be related to difference in cellular lifespans. If GL3 inclusions are less amenable to be broken down by α-Gal A, then cellular turnover may in fact play a major role in the observed clearance following ERT. Thus, ERT may in fact prevent the accumulation of new GL3 rather than clearing what has already accumulated inside the cells.

Our study had several limitations. Only two *GLA* mutations (*c.666delC,* a frameshift mutation; and *c.658C > T,* a nonsense mutation) were represented in our study, both of which had been reported to cause classic Fabry phenotype [33,34]. The number of patients was limited with M/F = 3/3. Although random X-inactivation could have influenced the severity of the symptoms, the overall spectrum of gastrointestinal and cardiac manifestations and responses to ERT were so uniform among our patients that we believe the conclusions were not significantly impacted by the gender heterogeneity.

## 6. Conclusions

The study suggests a different impact of ERT on human tissues affected by classic FD. A total of 0.2 mg/kg EOW of agalsidase-α for 2 years clears intestinal enterocytes from GL-3 inclusions, paralleled by improvement of gastrointestinal symptoms. In contrast, cardiomyocytes remain hypertrophic and retain their GL-3 content. This divergent response may be related to different cellular turnovers and different expression of IGF-II-R, which is the major intracellular carrier of α-Gal A.

## Figures and Tables

**Figure 1 jcm-11-01344-f001:**
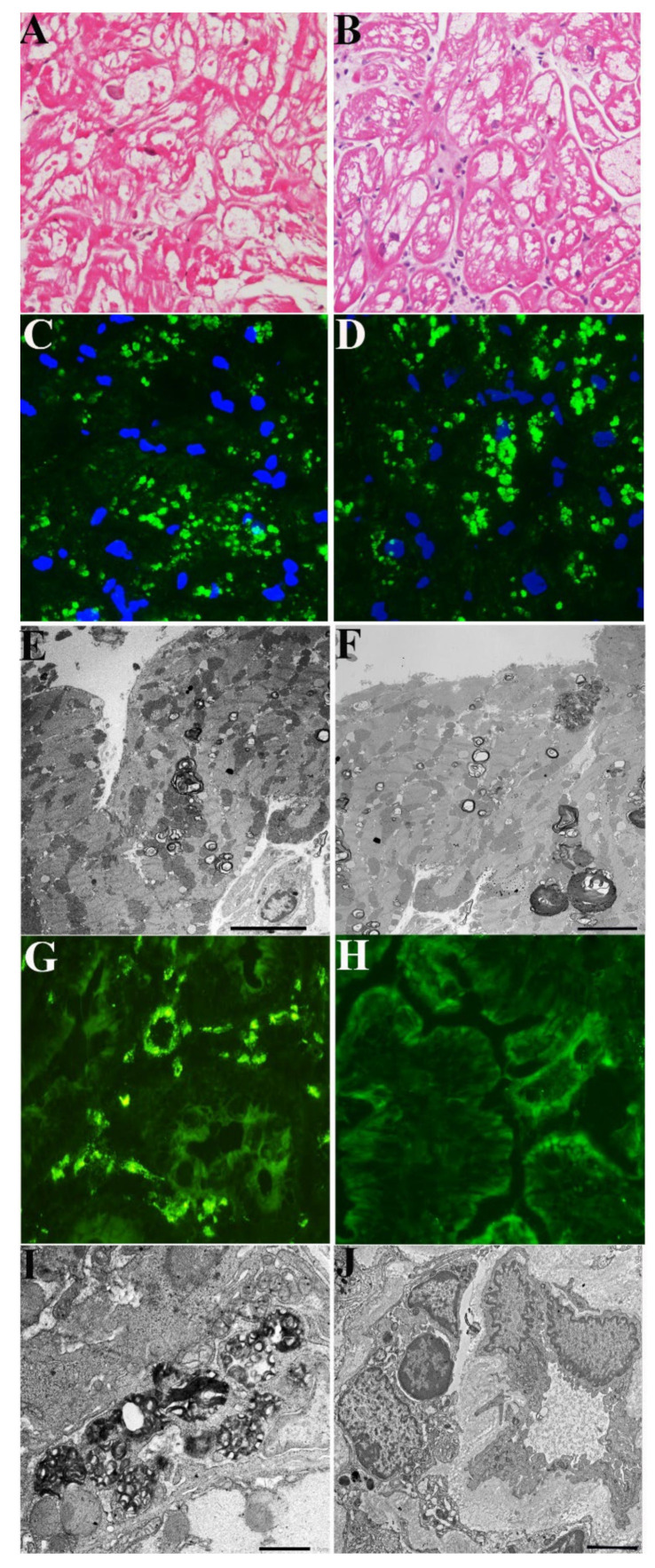
Pathologic evaluation of myocardial and intestinal tissue in a 72-year-old woman (Case 1) affected by Fabry disease before and after 2 years of agalsidase-α. (**A**–**F**) Myocardium unchanged cardiomyocyte dimensions and vacuolar area in H&E (**A**,**B**), immunofluorescence for Gb3 (**C**,**D**) and electron microscopy (**E**,**F**) comparing pre (**A**,**C**,**E**) and post (**B**,**D**,**F**) findings. (**G**–**J**) Intestinal tissue mobilization of Gb3 in immunofluorescence (**G**,**H**) and ultrastructural examination (**I**,**J**), comparing pre- (**G**,**I**) and post-treatment (**H**–**J**) features.

**Figure 2 jcm-11-01344-f002:**
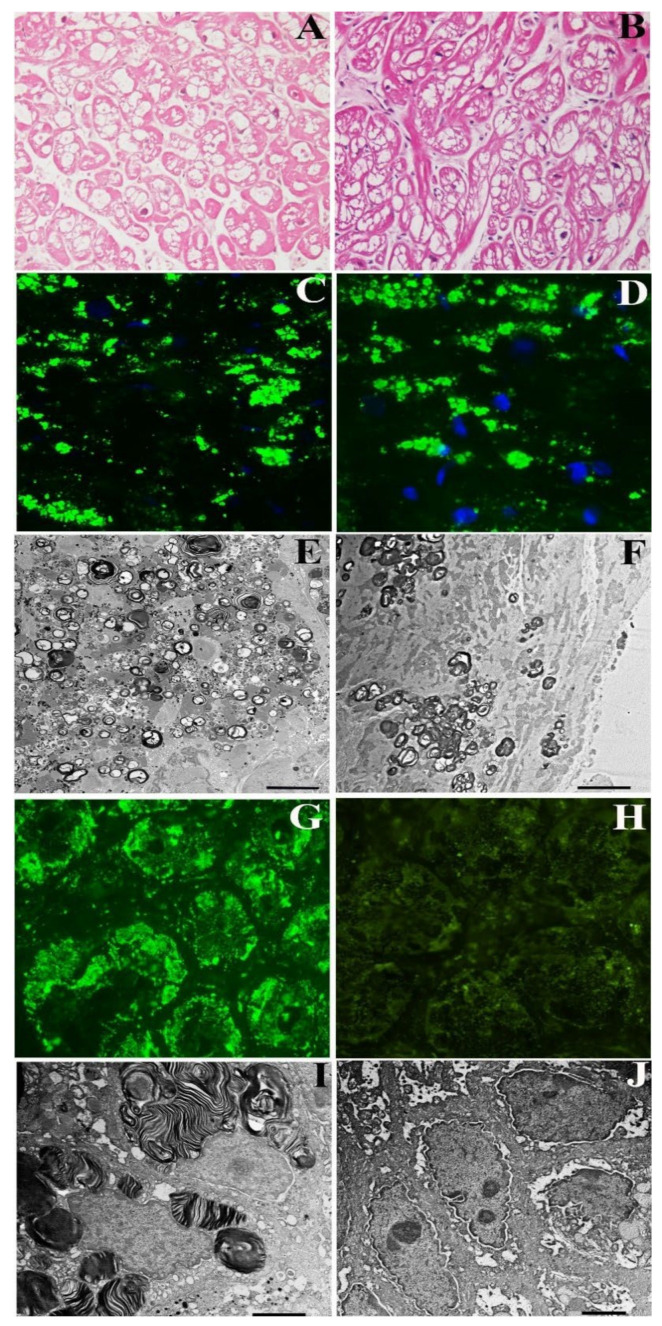
Pathologic evaluation of myocardial and intestinal tissue in a 54-year-old woman (Case 2) affected by Fabry disease before and after 2 years of agalsidase-α. (**A**–**F**) Myocardial tissue pre- (**A**,**C**,**E**) and post-treatment (**B**,**D**,**F**). There was no significant change in cardiomyocyte size or vacuolar area on H&E stains of pre-ERT (**A**) vs. post-ERT (**B**) biopsies. Likewise, there was no obvious reduction in GL3 immunofluorescence staining (**C**,**D**), and GL3 inclusions were not cleared from cardiomyocytes by electron microscopy (**E**,**D**) following treatment. (**G**,**H**) Complete clearance of GL3 in the intestinal tissue by immunofluorescence (**G**,**H**) and electron microscopy (**I**,**J**).

**Figure 3 jcm-11-01344-f003:**
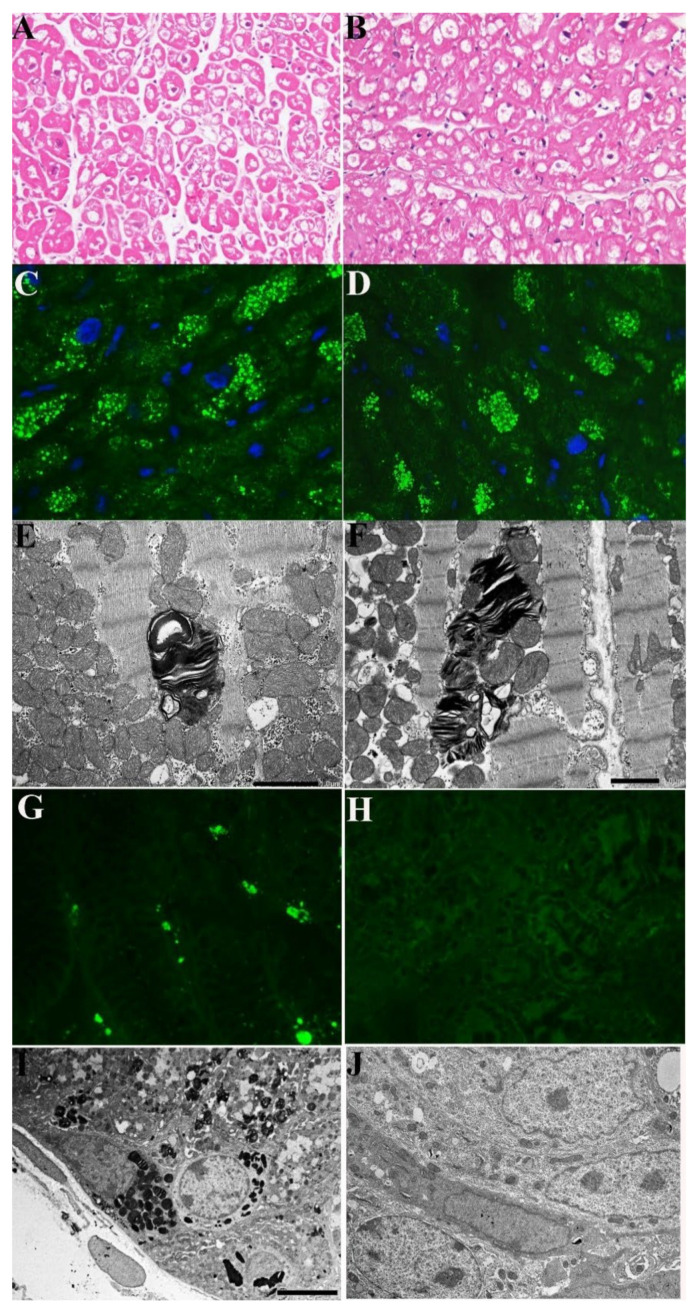
Pathologic evaluation of myocardial and intestinal tissue in a 43-year-old man (Case 3) affected by Fabry disease before and after 2 years of agalsidase-α. (**A**–**F**) No significant change in cardiomyocyte diameter and cytoplasmic vacuolar areas on H&E stains (**A**,**B**), GL3 accumulation by immunofluorescence microscopy (**C**,**D**) and ultrastructural examination (**E**,**F**). (**A**,**C**,**E**) Pre and panels. (**B**,**D**,**F**) Post-treatment. (**G**,**H**) Clearance of intestinal tissue from GL3 by immunofluorescence microscopy (**G**,**H**) and ultrastructural examination (**I**,**J**). (**G**,**I**) Pre and panels. (**H**,**J**) Post-treatment.

**Figure 4 jcm-11-01344-f004:**
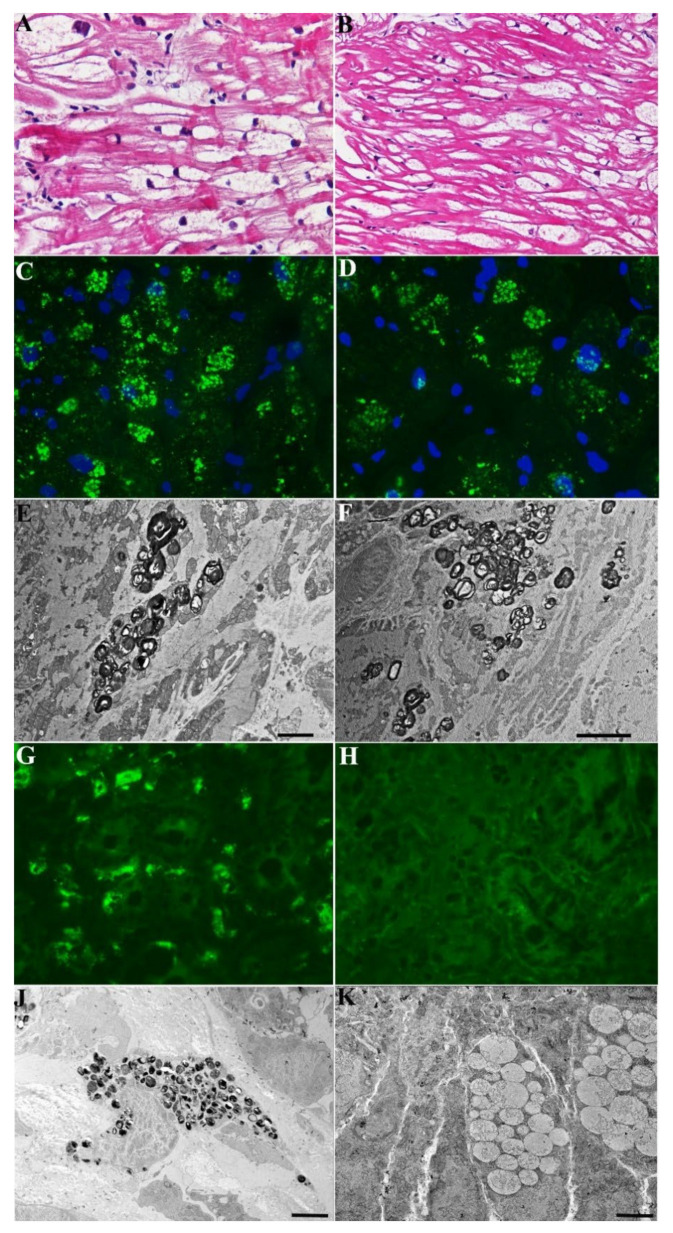
Pathologic evaluation of myocardial tissue (in pre-hypertrophic cardiomyopathy) and intestinal tissue (with diarrhea and malabsorption) in a 29-year-old man (Case 4) affected by Fab-ry disease before and after 2 years of agalsidase-α. (**A**–**F**) show in the myocardium unchanged cardiomyocyte dimensions and vacuolar area in H&E (**A**,**B**), immunofluorescence for GL3 (**C**,**D**) and electron-microscopy (**E**,**F**) comparing pre (**A**,**C**,**E**) and post (**B**,**D**,**F**) findings. Panel (**G**–**J**) shows in the intestinal tissue mobilization of Gb3 in immunofluorescence (**G**,**H**) and ultrastructur-al examination (**J**,**K**), comparing pre (**G**,**J**) and post–treatment (**H**,**K**) features.

**Figure 5 jcm-11-01344-f005:**
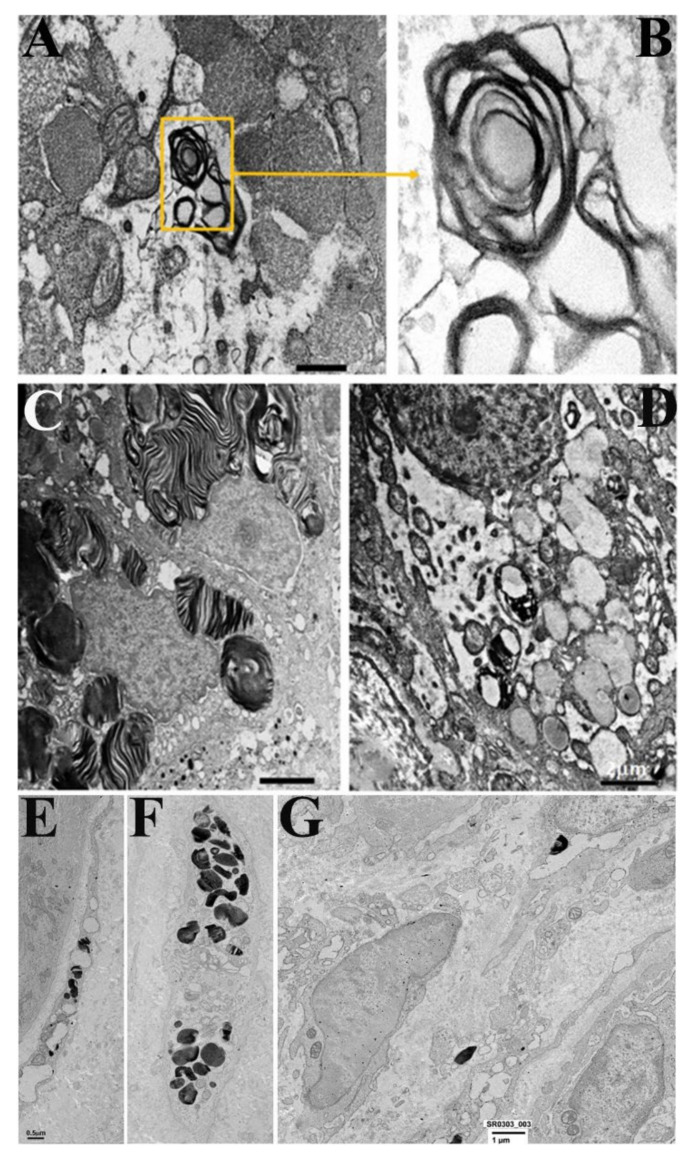
Secretory epithelium and fibroblasts of the gastric mucosa before and after ERT. (**A**,**B**) Secretory epithelial cell containing a glycolipid body (**A**) and with its detail at higher magnification. (**C**) Massive accumulation of glycolipid bodies. (**D**) Low-power micrographs. After ERT glycolipid bodies are absent. (**E**,**F**) Glycolipid bodies are visible in the cytoplasm of two fibroblasts. (G) Low magnification of the subepithelial region of the gastric mucosa. Glycolipid bodies are absent.

**Figure 6 jcm-11-01344-f006:**
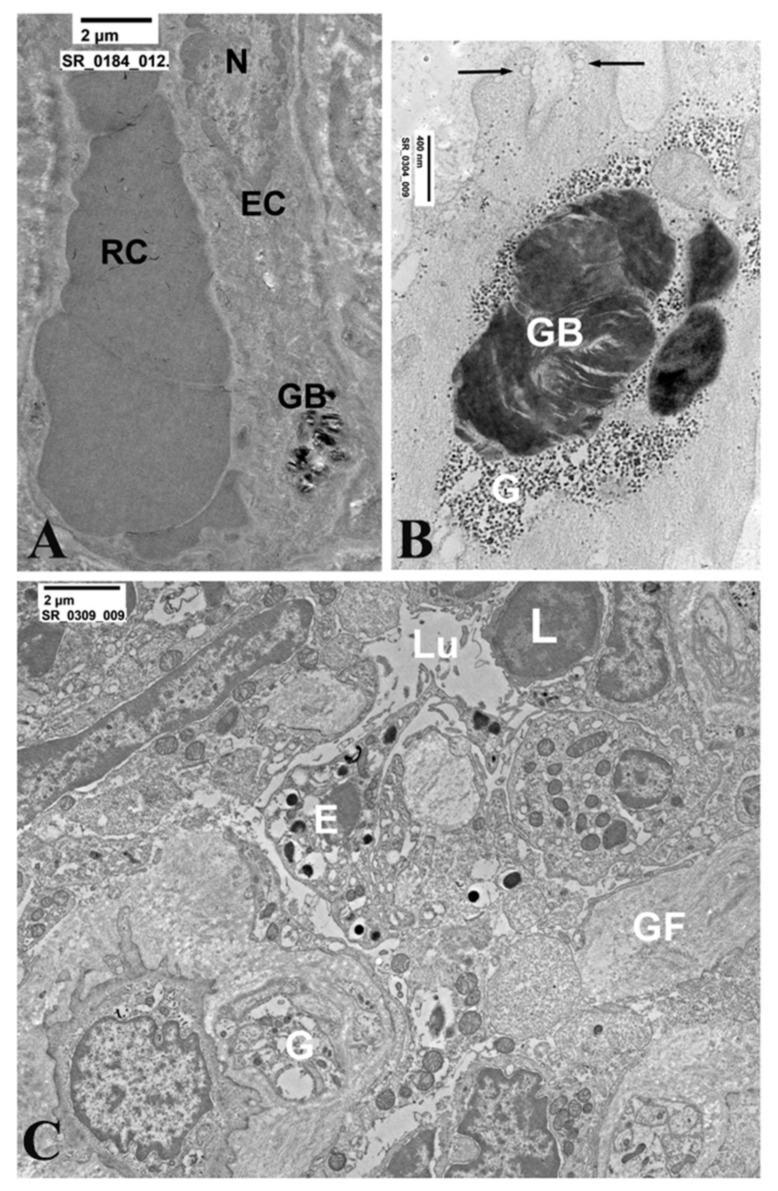
Intestinal cells before and after ERT. (**A**,**B**) Endothelial cell (EC) with glycolipid body (GB). N = nucleus of endothelial cell ERT. (**B**) Submucosal smooth muscle cell containing a large glycolipid body (GB) before ERT. Smooth muscle cell was recognized by the presence in the cytosol of contractile filaments and b-particles of glycogen (G) and by endocytotic vesicels on the plasmamembrane (arrows). (**C**) Subepitelial region showing different cell types, all devoid of glycolipid bodies after ERT.Lu = lumen of a vessel containing a lymphocyte (L). Gangliar longitudinal (GF) and transverse (GT) nervous processes. In the center, a type I endocrine cell is present, unaffected by glycolipid bodies.

**Figure 7 jcm-11-01344-f007:**
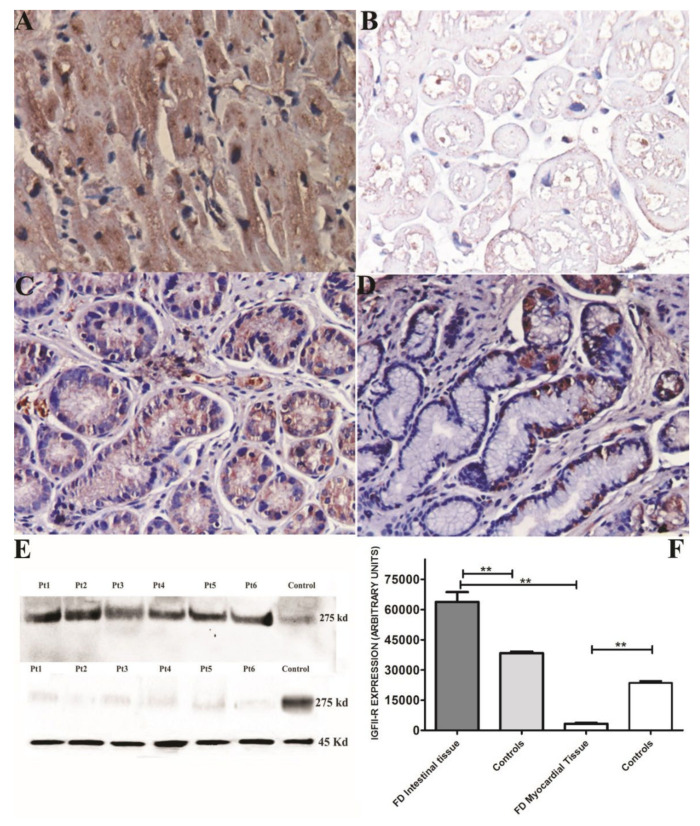
Assessment of IHC and Western blot quantification of IGF-II-R in the endomyocardial and intestinal biopsies in Fabry disease patients. (**A**–**D**) Ihc for IGF-II-R showing expression of the mannose-6-phosphate receptor in normal heart (**A**) compared with FD myocardial tissue (**B**) and in control intestinal tissue versus FD intestine. They show a remarkable reduction of myocardial receptors compared with intestinal ones and normal controls. (**E**) Graph showing expression of IGF-II-R in intestinal, myocardial tissue and normal control. In FD, heart expression was 19-fold lower than in FD intestine (*p* < 0.001). (**F**) Western blot of IGF-II-R in intestine and myocardium of FD patients and controls. Alpha sarcomeric actin (45 kDa) was used as the loading control. **: statistical significance value= *p* < 0.01.

**Table 1 jcm-11-01344-t001:** Clinical and pathologic data from 6 members belonging to two different families affected by myocardial and intestinal FD before and after 2 years of alpha-agalsidase treatment.

	Patient 1	Patient 2	Patient 3	Patient 4	Patient 5	Patient 6
Age/sex	72/F	54/F	43/M	29/M	21/F	35/M
Manifestation						
Pre	A, Dr, P	A, Dr, D	A, Dr, P, M	A, Dr, P, M, D	P, Dr, A	P, A, Dr
Post	P A, Dr, resolved	D A, Dr, resolved	P A, Dr, M resolved	D, P A, Dr, M, resolved	P Dr, A resolved	P A, Dr, resolved
Gene studies						
Gene mutation	c.666delC	c.666delC	c.666delC	c.666delC	c.666delC	c.658C > T
Lyso-Gb3 Range (0.8–1.13 nMol/L)	2.43	8.4	69.22	85.32	3.28	3.4
Enzymatic activity (nmol/h/mL)	4.2	3.1	0	0	6	0
CMR study						
LV mass (g) Mass/BSA						
Pre	114.12	118	172.1	126.65	47.80	120
Post	116	120	172	125	50	124
LVEF(%)						
Pre	54	56	58	60	64	59
Post	53	55	58	60	62	58
Cardiac Histologic Findings						
Cardiomyocyte diameter (µm)						
Pre	22.64 ± 4.41	24.91 ± 8.24	32.18 ± 7.43	21.63 ± 3.01	25.51 ± 6.46	28.18 ± 4.5
Post	26.72 ± 7.85	26.30 ± 4.62	35.14 ± 9.3	23.75 ± 2.87	28.87 ± 8.31	29.24 ±2.4
Extent of cell vacuoles (%)						
Pre	22%	30%	50%	27%	12%	32%
Post	24%	41%	48%	27%	14%	33%

Note: dyspnea = D, palpitation = P, abdominal pain = A, diarrhea = Dr, malabsorption = M.

## Data Availability

The datasets used and analyzed during the current study are available from the corresponding author upon reasonable request.

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
