# Peer review of "Divergent Impact of Enzyme Replacement Therapy on Human Cardiomyocytes and Enterocytes Affected by Fabry Disease: Correlation with Mannose-6-phosphate Receptor Expression"

_jcm, 2022, doi:10.3390/jcm11051344_

Round 1
Reviewer 1 Report
The study by Frustaci et al. showing different impact of ERT on cardiomyocytes and enterocytes in Fabry disease nicely illustrate clinical and histological features before and after 2 years of ERT. Given FD is not a common disease, this observational study design is acceptable, and the results seem to be clear to understand. There remain several concerns:
1. Materials and Methods:
Please show the study period of the present study. And show how many patients with FD were treated in that period. This information may give the readers how rare FD is in this area.
2. Table 1 CMR study:
Please add the information about LVEF because it is very important in management of cardiac involvement.
3. Figures 1 to 7:
How do the authors assess the possibility of sampling error of biopsy? The authors may have selected only the best pieces to show the difference. How do the authors say these pictures are representative of the tissue?
4. Case studies, Case 3:
The authors seem to diagnose iron deficiency anemia by the levels of serum iron. However, it is usually diagnosed by serum ferritin.
5. Discussion:
Reference No. 29 shows reduction in myocardial GL3 but it is not the case in the present study. Is there any difference of effect of ERT according to each mutation?
6. Discussion: line 455
ERT may reduce cardiac events. If ERT does not clear GL3 from cardiomyocyte, why was the events reduced? Please discuss it.
7. Discussion: line 492
VEBs were reduced. If ERT does not clear GL3 from cardiomyocyte, why were VEBs reduced? Please discuss it. (Please spell out VEBs.)
8. Discussion: line 496
"Podocytes and arterial smooth muscle cells are more resistant to ERT compared with endothelial cells, mesangial cells or fibroblasts in the kidney. Arterial smooth muscle cells in the skin and cardiomyocytes in the heart show more resistance to ERT compared to endothelial cells in the same tissues." These statements seem to be based on previous studies but not the present study. Please indicate which study shows the results.
Author Response
The study by Frustaci et al. showing different impact of ERT on cardiomyocytes and enterocytes in Fabry disease nicely illustrate clinical and histological features before and after 2 years of ERT. Given FD is not a common disease, this observational study design is acceptable, and the results seem to be clear to understand. There remain several concerns:
- Materials and Methods:
Please show the study period of the present study. And show how many patients with FD were treated in that period. This information may give the readers how rare FD is in this area.
Reply: 88 FD pts were treated during the study.
- Table 1 CMR study:
Please add the information about LVEF because it is very important in management of cardiac involvement.
Reply: CMR informations regarding LVEF before and after ERT have been added in Table 1.
- Figures 1 to 7:
How do the authors assess the possibility of sampling error of biopsy? The authors may have selected only the best pieces to show the difference. How do the authors say these pictures are representative of the tissue?
Reply: We agree that sampling is a main limitation of endomyocardial biopsy. We have been convinced on representativity of the biopsies from the evidence that all samples examined contained similar changes.
- Case studies, Case 3:
The authors seem to diagnose iron deficiency anemia by the levels of serum iron. However, it is usually diagnosed by serum ferritin.
Reply: Thank you for the specification. Ferritin was in all patients below the normal values
(< 15 ng/ml; NV 15-300). This is specified in the text.
- Discussion:
Reference No. 29 shows reduction in myocardial GL3 but it is not the case in the present study. Is there any difference of effect of ERT according to each mutation?
Reply: No difference on the ERT effect was observed according with each mutation. Morover in the discussion we underline that the first study from Hughes reporting a 20% reduction of Gb3 following ERT (Reference 29) was not confirmed by further studies (see Thurnberg et al).
- Discussion: line 455
ERT may reduce cardiac events. If ERT does not clear GL3 from cardiomyocyte, why was the events reduced? Please discuss it.
Reply: The point raised is really interesting and we have not at the moment a clearcut explanation . We can speculate that reduction on the interstitial amount of Gb3, particularly those around conduction tissue, might have had a role. This point is added in the discussion.
- Discussion: line 492
VEBs were reduced. If ERT does not clear GL3 from cardiomyocyte, why were VEBs reduced? Please discuss it. (Please spell out VEBs.)
Reply: The point raised is really interesting and we have not at the moment a clearcut explanation . We can speculate that reduction on the interstitial amount of Gb3 might have had a role. This point is added in the discussion.
- Discussion: line 496
"Podocytes and arterial smooth muscle cells are more resistant to ERT compared with endothelial cells, mesangial cells or fibroblasts in the kidney. Arterial smooth muscle cells in the skin and cardiomyocytes in the heart show more resistance to ERT compared to endothelial cells in the same tissues." These statements seem to be based on previous studies but not the present study. Please indicate which study shows the results.
Reply: Appropriate reference (n 33) has been added.
Reviewer 2 Report
The authors speculate that the divergent impact of enzyme replacement therapy with agalsidase alpha on different human tissues may in part be explained by expression of mannose-6-phosphate receptors. This is a small study of only six patients but is notable for study of biopsy materialThere are indeed few studies which have utilised such biopsy material, the Hughes et al being the first. The abstract can be improved for use of concise language. The abstract and line 121 should specify that the efficacy of ERT is dependent on cellular uptake which in turn is dependent upon mannose-6 phosphate receptor expression. The reference in line 61 is missing; lines 159 and 160 have typing errors .
A notable finding is that mannose-6-phosphate receptor expression in cardiac tissue from Fabry disease was sevenfold lower than in normal cardiac tissue. Was this already known? Further discussion is warranted and the authors could discuss the value of chaperone therapy as an alternative approach to cardiac Fabry disease in this regard. The mutation c.666delC is not amenable to migalastat. The impact of enzyme replacement therapy on inflammation warrants mention. Was it measured?
Author Response
The authors speculate that the divergent impact of enzyme replacement therapy with agalsidase alpha on different human tissues may in part be explained by expression of mannose-6-phosphate receptors. This is a small study of only six patients but is notable for study of biopsy material
There are indeed few studies which have utilised such biopsy material, the Hughes et al being the first.
The abstract can be improved for use of concise language. The abstract and line 121 should specify that the efficacy of ERT is dependent on cellular uptake which in turn is dependent upon mannose-6 phosphate receptor expression. The reference in line 61 is missing; lines 159 and 160 have typing errors .
Reply: Many thanks for your observations. Abstract and discussion have been implemented with your appropriate suggestions. The reference 31 has been mentioned in the text.
A notable finding is that mannose-6-phosphate receptor expression in cardiac tissue from Fabry disease was sevenfold lower than in normal cardiac tissue. Was this already known? Further discussion is warranted and the authors could discuss the value of chaperone therapy as an alternative approach to cardiac Fabry disease in this regard. The mutation c.666delC is not amenable to migalastat.
The impact of enzyme replacement therapy on inflammation warrants mention. Was it measured?
Reply: We thank the reviewer for his/her observation. Clear evidence of myocardial inflammation (i.e. ≥ 14 leukocytes /2mm associated to necrosis of adjacent myocytes) were not documented in the histologic sections of our patient population. We agree that this is the first report on assessment of Mannose-6-phosphate receptors on FD tissue.
This manuscript is a resubmission of an earlier submission. The following is a list of the peer review reports and author responses from that submission.
Round 1
Reviewer 1 Report
In this work, the authors show the differential curative effect of the ERT on cardiac versus intestinal tissues of 6 patients affected by Fabry disease. Specifically, the authors show the success of ERT on intestinal but not in cardiac tissues and provide evidence that such divergent effects could be linked to the different protein expression levels of the mannose-6-phosphate receptor which is primarily involved in the uptake of the exogenous enzyme and its delivery to the lysosomal lumen.
Although the clear clinical relevance of the work in the lysosomal storage disease therapy field, the presented manuscript needs extensive editing towards a canonical and regular style of a scientific journal. For instance, the figure legends must report sufficient information regarding the objective description of the experiment reporting a concise summary of the method used without reporting any conclusion.
Moreover, in order to explain the divergent impact of the ERT on cardiomyocytes versus enterocytes, the authors looked at the different expression levels of the M6P receptor in the two groups of tissues. However, since the sole quote of the receptor exposed on the plasma membrane is actually responsible for the uptake of the exogenous enzyme, the authors must show this instead of the total protein levels to support such conclusion and correlation.